# Late Blight Resistance Conferred by *Rpi-Smira2/R8* in Potato Genotypes In Vitro Depends on the Genetic Background

**DOI:** 10.3390/plants11101319

**Published:** 2022-05-16

**Authors:** Eva Blatnik, Marinka Horvat, Sabina Berne, Miha Humar, Peter Dolničar, Vladimir Meglič

**Affiliations:** 1Crop Science Department, Agricultural Institute of Slovenia, Hacquetova ulica 17, SI-1000 Ljubljana, Slovenia; peter.dolnicar@kis.si (P.D.); vladimir.meglic@kis.si (V.M.); 2Department of Agronomy, Biotechnical Faculty, University of Ljubljana, Jamnikarjeva ulica 101, SI-1000 Ljubljana, Slovenia; marinka.horvat@ijs.si (M.H.); sabina.berne@bf.uni-lj.si (S.B.); 3Department of Wood Science and Technology, Biotechnical Faculty, University of Ljubljana, Jamnikarjeva ulica 101, SI-1000 Ljubljana, Slovenia; miha.humar@bf.uni-lj.si

**Keywords:** potato *R8* genotypes, Sárpo Mira, *Rpi-Smira2/R8*, *Phytophthora infestans*, in vitro infection, *Solanum tuberosum*

## Abstract

Potato production worldwide is threatened by late blight, caused by the oomycete *Phytophthora infestans* (Mont.) de Bary. Highly resistant potato cultivars were developed in breeding programs, using resistance gene pyramiding methods. In Sárpo Mira potatoes, five resistance genes (*R3a*, *R3b*, *R4*, *Rpi-Smira1*, and *Rpi*-*Smira2/R8*) are reported, with the latter gene assumed to be the major contributor. To study the level of late blight resistance conferred by the *Rpi-Smira2/R8* gene, potato genotypes with only the *Rpi-Smira2/R8* gene were selected from progeny population in which susceptible cultivars were crossed with Sárpo Mira. Ten *R8* potato genotypes were obtained using stepwise marker-assisted selection, and agroinfiltration of the avirulence effector gene *Avr4*. Nine of these *R8* genotypes were infected with both Slovenian *P. infestans* isolates and aggressive foreign isolates. All the progeny *R8* genotypes are resistant to the Slovenian *P. infestans* isolate 02_07, and several show milder late blight symptoms than the corresponding susceptible parent after inoculation with other isolates. When inoculated with foreign *P. infestans* isolates, the genotype C571 shows intermediate resistance, similar to that of Sárpo Mira. These results suggest that *Rpi-Smira2/R8* contributes to late blight resistance, although this resistance is not guaranteed solely by the presence of the *R8* in the genome.

## 1. Introduction

The potato (*Solanum tuberosum* L.) holds a significant place in global crop production, with over 370 million tons produced in 2019 [1]. Although the potato was not immediately accepted by European farmers after its first introduction in the late 16th century by Spanish colonizers, nowadays potatoes remain an essential staple food crop in Europe, with over 55.3 million tons harvested in 2020 [2].

In particular, late blight, caused by the oomycete *Phytophthora infestans* (Mont.) de Bary, is considered one of the most devastating plant diseases, and poses a challenge to both organic and conventional potato production systems worldwide [3]. It is estimated that late blight causes up to EUR 1 billion losses in costs with the control and reduction of production [4]. Fungicides are most frequently used in late blight control, however, their mass application increases the evolutionary pressure on *P. infestans*, which could potentially lead to the development of fungicide resistance [5].

*Phytophthora infestans* is a hemibiotroph, meaning that its life cycle is both biotrophic and necrotrophic [6]. This pathogen can reproduce asexually or sexually, depending on the presence of the two mating types, referred to as A1 and A2 [7]. In asexual reproduction, sporangia are carried either by wind or water to their host cells, where they germinate directly, by forming invasive hyphae, or indirectly, by zoospores [8].

Until the 1980s, only isolates of *P. infestans* of mating type A1 existed in Europe, which prevented *P. infestans* from increasing its genetic diversity through sexual recombination [9]. In the period between 1980 and 1988, several European countries began to report the appearance of the isolates of the A2 mating type [10]. This drastically changed the *P. infestans* populations, especially in north-western Europe, leading to fungicide resistance, higher environmental adaptability, and disease control challenges [11]. In 2004, the genotype EU_13_A2 (also known as Blue13) spread out across north-western Europe, and became the dominant genotype in European fields for five years [12]. Its aggressiveness and resistance to phenylamide fungicides made it difficult for farmers to control the disease [13]. In 2017, the presence of Blue13 decreased, replaced by other adapted genotypes, such as EU_36_A2, EU_37_A2, and EU_41_A2, which still continue to spread today [14].

In addition to late blight research and control of the disease using fungicides, potato breeding programs continuously developed new potato cultivars with high resistance to late blight [4,15,16,17,18,19]. Plant resistance is divided into two types: vertical resistance (race-specific) and horizontal (race-non-specific, partial resistance) [20]. Vertical resistance involves the major resistance (*R*) genes that produce R proteins, which interact with the effector proteins secreted by *P. infestans*. This interaction requires the presence of both products of the *R* gene, and a corresponding avirulence effector gene (*Avr*) from the pathogen to result in plant resistance, according to the gene-for-gene model [21]. Pathogen effectors bind to the complementary R proteins in plants, resulting in the hypersensitive response as part of the effector-triggered immunity (ETI) [8,22,23,24]. Based on the same model, the transient agroinfiltration of the effector genes is used in many potato studies for gene function analysis [25], transgenic potato plant production [26], metabolic studies [27,28], and resistance enhancement [29,30].

In the mid-20th century, several wild potato species carrying major *R* genes were identified, mostly *Solanum demissum,* with eleven race-specific *R* genes (*R1*–*R11*) [31]. Some of these *R* genes were introduced into potatoes to form new resistant cultivars with multiple stacked *R* genes (gene pyramiding), which proved to have stronger and more durable resistance [20]. As the *P. infestans* population developed and evolved into more complex races and highly adaptable strains, these cultivars were soon overcome by the pathogen, and the major *R* genes derived from *S. demissum* were no longer sufficient [32]. Researchers focused on horizontal or quantitative resistance, conferred by QTLs, and discovered new sources of resistance from wild potato species. To date, more than 60 *Rpi* and *R* genes have been identified, the most recent being *Rpi-amr1*, which confers broad-spectrum resistance to late blight [33,34].

In the 1990s, a potato breeding program, led by the Sárvari family, developed a highly late blight-resistant cultivar, Sárpo Mira [35]. It was trialed in the United Kingdom, and included in the United Kingdom National List in 2002 [36]. After almost three decades, Sárpo Mira is still resistant to late blight, even after inoculation with the aggressive isolate EU_13_A2 [37,38]. The exact molecular mechanisms underlying this resistance in Sárpo Mira are still under investigation. It is known that resistance is conferred by four qualitative resistance genes, *R3a*, *R3b*, *R4*, and *Rpi-Smira1*, and one quantitative gene, *Rpi-Smira2* [39]. However, Sárpo Mira resistance differs depending on whether the whole plant was infected, or whether only detached leaves were inoculated [40,41,42].

Of the five *R* genes in Sárpo Mira, *Rpi-Smira1* and *Rpi-Smira2* are the subject of molecular and resistance studies [36,43,44,45,46,47]. *Rpi-Smira1* is located on chromosome XI, near the *R3* gene cluster of the potato genome, which is associated with the location of several other *S. demissum*-originating resistance genes [36]. In a PhD dissertation by Jo [48], it is suggested that *Rpi-Smira2* and *R8* are identical, or functional homologs, and confer similar resistance levels. Genetic mapping reveals that the *R8* gene is located on the long arm of chromosome IX [46], and not on the short arm of chromosome XI, as originally hypothesized [49]. The *R8* gene is also present in Mastenbroek *R8* (*MaR8*) differential plants, the cultivars Jacqueline Lee and Missaukee, and the Chinese cultivars PB-06 and S-60, shown to have similar late blight resistance levels to Sárpo Mira in field trials [50]. Therefore, the *Rpi-Smira2/R8* gene was hypothesized to be the predominant source of *P. infestans* resistance in Sárpo Mira. So far, late blight resistance studies of the *Rpi-Smira2/R8* gene include either transgenic plants with the *R8* gene [50], which limits its use in potato breeding programs, or the F1 progeny from *MaR8* differential plants containing only the *R8* gene not originating from Sárpo Mira [51]. The F1 progeny from Sárpo Mira shows intermediate or high late blight resistance; however those progeny genotypes were not analyzed with genetic markers for the presence of the five Sárpo Mira *R* genes [39]. Therefore, this late blight resistance could not be attributed to a single *R* gene.

In potato resistance research, in vitro techniques are mostly used to propagate plant material under sterile and controlled conditions, and to rapidly generate numerous plantlets for infection studies in a greenhouse environment [31,42,52]. However, few studies use potato plantlets for direct inoculation of pathogens and observation of disease progression in vitro [53,54,55,56]. 

In this study, we: (i) applied marker-assisted selection (MAS) and effector agroinfiltration to select potato genotypes containing only *Rpi-Smira2/R8* from the progeny of susceptible cultivars and Sárpo Mira, and (ii) conducted in vitro inoculation assays of these selected potato genotypes with *P. infestans* isolates, differing in geographical origin and aggressiveness, to compare them with the corresponding susceptible and resistant parental cultivars. Our objective was to determine the contribution of the *Rpi-Smira2/R8* gene, originating from Sárpo Mira, to late blight resistance in these progeny potato genotypes, while ensuring uniform plant material and a contamination-free tissue culture environment.

## 2. Results

### 2.1. Marker-Assisted Selection and Avr4 efector Agroinfiltration of Progeny Potato Genotypes Carrying Rpi-Smira2/R8 Gene

To evaluate the contribution of the *Rpi-Smira2/R8* gene, originating from Sárpo Mira, to late blight resistance in potato, we first obtained plant genotypes carrying only the *Rpi-Smira2/R8* gene, and none of the other four *R* genes from Sárpo Mira. Therefore, 1420 progeny seeds of crosses between Sárpo Mira and five susceptible cultivars (Rioja, Lusa, Bikini, Colomba, and Sylvana) were sown in the greenhouse, and three rounds of selection with genetic markers were performed on 1213 successfully germinated progeny plants (Figure 1). Only 186 samples lacked the *R3b* gene, and we then proceeded onto negative selection for genetic markers for *R3a* and *Rpi-Smira1*, resulting in 104 samples. The final genetic selection was performed using the genetic marker R8-184, where 36 samples were positive for the *R8* gene. At this point, only progeny from four of five crosses completed this selection, as no genotypes were obtained from the cross of Bikini and Sárpo Mira. To the best of our knowledge, no genetic marker for *R4* has yet been developed, so agroinfiltration was used as an alternative selection approach.

In this study, effector agroinfiltration of the *Avr4* gene was performed on detached leaves of 36 genotypes positive for the *R8* gene, along with Sárpo Mira as a positive control. Detached leaves were carefully nicked on the surface of the abaxial side with needles under a stereomicroscope to successfully infiltrate the bacterial suspension with a syringe. At least four sites per leaflet were nicked several times, which was approximately the diameter of a syringe, and three leaflets were used for infiltration. A total of three leaves per genotype were used to include the mock and negative controls. Three days after infiltration, the leaves were examined for the presence or absence of a hypersensitive response (Figure 2). Of the 36 tested genotypes, 17 show signs of hypersensitive response indicating the *R4* gene presence, and 5 genotypes return inconclusive results and are, therefore, excluded from the final collection. 

The final selection yielded ten *R8* potato plants (0.82% of all germinated progeny) that contained only the *Rpi-Smira2/R8* gene among the five *R* genes of Sárpo Mira. These were two genotypes from the Rioja cross (R7, R15), one genotype from the Lusa cross (L166), three genotypes from the Colomba cross (C419, C557, and C571), and four genotypes from the Sylvana cross (S859, S989, S999, and S1219). All ten *R8* genotypes and parental cultivars were tested for the four resistance genes (*R3a*, *R3b*, *Rpi-Smira1*, and *Rpi-Smira2/R8*) to confirm the presence of the *Rpi-Smira2/R8* gene, and the absence of other *R* genes in the progeny plants (Figure 3). Rioja is positive for the *R3b* (Figure 3a) and *R3a* genes (Figure 3b), whereas Lusa and Colomba are both negative. Sylvana is positive for the *R3a* gene (Figure 3b). All four susceptible cultivars are negative for the *Rpi-Smira1* (Figure 3c) and *Rpi-Smira2/R8* genes (Figure 3d), while Sárpo Mira is positive for all four resistance genes tested.

### 2.2. Disease Progression of Phytophthora infestans Isolates

Four *P. infestans* isolates of different origins, which also differed in their growth after a successful infection, were used in this study. The isolates 90128 and 02_07 begin to sporulate after covering a larger leaflet area with mycelia, although 02_07 rarely reaches this growth stage, whereas the isolates IPO-C and 09_07 sporulate almost immediately after mycelial development with numerous sporangia.

The area under the disease progression curve (AUDPC) value was calculated for each *P. infestans* isolate across all *R8* genotypes and parental cultivars, to compare disease progression and aggressiveness (Figure 4). IPO-C has the highest AUDPC value (23.7), whereas the AUDPC of 90128 is slightly lower (22.4), although the difference between these two isolates is not statistically significant (*p* > 0.05, Tukey’s range test). In comparison, the isolate 09_07 shows similar disease progression to the foreign isolates in terms of growth and sporulation, but the AUDPC of 09_07 (17.4) is statistically significantly different (*p* < 0.05) from the foreign isolates and 02_07, which has the lowest AUDPC (7.8). These results indicate that 02_07 is the least aggressive isolate used in this study, whereas 09_07 has an intermediate level of aggressiveness.

### 2.3. Late Blight Resistance of Progeny R8 Genotypes

A total of nine progeny *R8* genotypes were infected in vitro with four *P. infestans* isolates, along with the corresponding susceptible parental cultivar (Rioja, Lusa, Colomba, and Sylvana), and the resistant cultivar Sárpo Mira. The genotype S1219 was excluded from the experiments, due to its poor growth in tissue culture. Data obtained from all in vitro experiments were combined into four separate cross groups of the progeny *R8* genotypes, containing the corresponding susceptible parental cultivar and the resistant cultivar Sárpo Mira (Rioja, Lusa, Colomba, and Sylvana cross group). Each cross group was evaluated for susceptibility or resistance to each of the four *P. infestans* isolates (Table 1). Data were plotted for each cross group as a dot plot, with local polynomial regression fitting to compare the mean disease (MD) scores of the progeny *R8* genotypes for each isolate over an eight day period (Figure 5 and Appendix A).

In the Rioja cross group, the resistance level of two progeny *R8* genotypes (R7 and R15) are compared to susceptible Rioja and resistant Sárpo Mira. Infection with 02_07 causes similar mild disease symptoms in R7, R15, and Rioja, and late blight progresses slowly on plantlets throughout the inoculation experiment (Appendix A). The MD scores for R7, R15, and Rioja are 2.56, 2.13, and 3.28, respectively; therefore, all three are resistant to 02_07 (Table 1). This is not evident in the case of the isolate 09_07, where both progeny *R8* genotypes and Rioja are successfully infected by this isolate, and the severity of symptoms on the leaflets increases from 3 to 6 days post inoculation (dpi, after which it reached its stationary phase, showing logarithmic growth (Appendix A). Only the genotype R7 is susceptible to the isolate 09_07 (MD score of 6.00), while the genotype R15 and cultivar Rioja have an intermediate response (MD score of 5.75 and 5.83, respectively). However, all three are statistically significantly different from the resistant Sárpo Mira (*p* < 0.05), with an MD score of 1.00 (Table 1). After inoculation with the isolate 90128, the genotype R15 has the highest MD score of 6.50, while the genotype R7 has an MD score of 5.67, and are considered susceptible and intermediate, respectively. Rioja is also considered susceptible, due to its MD score of 6.00. For the isolate IPO-C, the genotype R7 has the highest MD score of 7.39, and is statistically significantly different from Sárpo Mira (*p* < 0.05), but not from Rioja. Both genotypes R7 and R15 are susceptible to IPO-C, while Rioja and Sarpo Mira have intermediate responses (Table 1). The disease progression curves for the isolates 90128 and IPO-C are not characteristic for logarithmic growth, but instead lean towards linear growth (Appendix A). This is the case for all genotypes and cultivars in the Rioja cross, except for the genotype R15 for the isolate 90128, where some data were missing at 3 dpi, due to a technical error.

The Lusa cross group consists of only one progeny *R8* genotype, L166, susceptible parental cultivar Lusa, and resistant Sárpo Mira. Although the genotype L166 has higher MD scores at 8 days post inoculation for the isolates 02_07, 09_07, and 90128 (3.33, 5.22, and 6.44, respectively) compared with the susceptible cultivar Lusa (1.72, 4.89, and 5.89, respectively), this difference is statistically significant only in the case of the isolate 02_07 (*p* < 0.05). Throughout the inoculation experiment with the isolates 02_07 and 09_07, the genotype L166 generally has higher MD scores compared to the susceptible cultivar Lusa, and its disease progression curve resembles a linear growth (Appendix A). In addition, the genotype L166 has statistically significantly higher MD scores for the isolates 02_07, 09_07, and 90128 compared with Sárpo Mira (*p* < 0.05). After inoculation with the isolates IPO-C and 90128, the disease progression curves for the genotype L166 and both parental cultivars also resemble a linear growth (Appendix A). In addition, a Tukey’s range test conducted at the 8 dpi shows that the genotype L166 is not statistically significantly different to any of the parental cultivars (Table 1).

All genotypes and parental cultivars in the Colomba cross group have low MD scores (ranging from 0.78 to 2.33) for the isolate 02_07 at 8 dpi (Table 1), and are therefore resistant to this isolate. In addition, similar to other cross groups, the disease progression curves for the isolate 02_07 show slow progression of late blight symptoms (Figure 5a). In the case of the isolate 09_07, all three progeny *R8* genotypes (C419, C557, and C571) have lower MD scores (4.33, 5.39, and 3.39 respectively) compared to the susceptible parental cultivar Colomba (MD score of 6.33), but higher compared to the resistant cultivar Sárpo Mira (Table 1). In addition, all progeny genotypes and the parental cultivar Colomba show logarithmic spread of late blight symptoms after inoculation with the isolate 09_07 (Figure 5b). Although the genotypes C419 and C557 show intermediate disease response, they are not statistically significantly different from the susceptible Colomba (*p* > 0.05). However, they are statistically significantly different to Sárpo Mira (*p* < 0.05). The genotype C571 is the only progeny genotype with an MD score at 8 dpi for the isolate 09_07 that is statistically significantly different from Colomba (*p* < 0.05), while showing a similar resistant response to that of Sárpo Mira, where there is no statistically significant difference (*p* > 0.05). After inoculation with the isolate 90128, both of the genotypes C419 and C571 (MD scores of 5.72 and 5.22, respectively, at 8 dpi) show an intermediate response in contrast to Colomba (MD score of 7.44), which has a susceptible response, and this difference is statistically significant (*p* < 0.05). Both genotypes show no statistically significant difference from Sárpo Mira (MD score of 4.56, *p* > 0.05), which also shows an intermediate response to the isolate 90128 (Table 1). Inoculation with the isolate IPO-C results in susceptible responses of the genotypes C419, C557, and Colomba (MD scores at 8 dpi of 7.22, 7.61, and 7.94 respectively), which is also statistically significantly different to the intermediate response of the genotype C571 and Sárpo Mira (MD scores of 5.61 and 5.33, respectively, *p* < 0.05). Tukey’s range test shows no statistically significant difference between C571 and Sárpo Mira (*p* > 0.05) (Table 1). The disease progression curves of the progeny *R8* genotypes and parental cultivars of Colomba cross group for the isolates 90128 and IPO-C are not entirely characteristic of logarithmic growth (Figure 5c,d).

Sylvana and its three progeny *R8* genotypes (S859, S989, and S999) show low MD scores for the isolate 02_07, ranging from 1.56 to 2.78. (Table 1), and none of them are statistically significantly different from Sárpo Mira (MD score of 0.94, *p* > 0.05), making them all resistant to 02_07. Similar to the other *R8* genotypes and parental cultivars, the disease progression curves of the Sylvana cross group for the isolate 02_07 show slow progression (Appendix A). The disease response of the genotypes S989 and S999 (MD scores of 4.56 and 4.11, respectively) to the isolate 09_07 are statistically significantly different from both the susceptible cultivar Sylvana (MD score of 7.17, *p* < 0.05), and the resistant Sárpo Mira (MD score of 1.00, *p* < 0.05), resulting in an intermediate response. The genotype S859 also has an intermediate response, with an MD score of 5.67, but is not statistically significantly different to either Sylvana or the genotypes S989 and S999 (*p* > 0.05). However, there is a statistically significant difference between the genotype S859 and Sárpo Mira (*p* < 0.05) (Table 1). The disease progression curves of the progeny *R8* genotypes and susceptible cultivar Sylvana, after inoculation with the isolate 09_07, follow logarithmic growth, similar to the other cross groups (Appendix A). For both the foreign isolates, 90128 and IPO-C, the MD scores of all three *R8* progeny genotypes (S859, S989, and S999) are not statistically significantly different from either the susceptible Sylvana (MD score of 7.00 for 90128, and 7.40 for IPO-C, *p* > 0.05) or the intermediate response of Sárpo Mira (MD score of 4.56 for 90128, and 5.33 for IPO-C, *p* > 0.05). All three *R8* genotypes show a susceptible response to the isolate IPO-C, whereas for the isolate 90128, only the genotype S999 shows a susceptible response, while the genotypes S859 and S989 show intermediate responses (Table 1). The disease progression curves of the Sylvana cross group for both the foreign isolates, 90128 and IPO-C, do not resemble logarithmic growth (Appendix A).

Sárpo Mira has the lowest MD scores for the two Slovenian isolates, 02_07 and 09_07, (0.94 and 1.00, respectively), and is, therefore, highly resistant to these two isolates (Table 1). In comparison, the MD scores for Sárpo Mira, after inoculation with the foreign isolates 90128 and IPO-C, are 4.56 and 5.33, respectively, indicating intermediate resistance.

## 3. Discussion

### 3.1. Combining Available PCR-Based Markers for Late Blight Resistance and Avr4 Effector Gene Agroinfiltration Enabled the Selection of Potato Plantlets Carrying Solely Rpi-Smira2/R8 Gene

Selection with genetic markers, or marker-assisted selection (MAS), is a reliable method used in breeding programs for rapid and accurate selection of potato genotypes with desirable traits, such as tuber flesh color [57], and resistance to pathogens [58]. In this study, our aim was to find specific genotypes from the whole progeny population that carried the *Rpi-Smira2/R8* gene, but were also lacking other *R* genes originating from Sárpo Mira (*R3a*, *R3b* and *Rpi-Smira1*), in order to use them in *P. infestans* resistance assays. 

To the best of our knowledge, no genetic marker is available for the *R4* gene, therefore, agroinfiltration of the *Avr4* effector gene was applied as an effective and established method [24]. In our study, we successfully modified the bacterial infiltration step by using detached potato leaves, instead of whole plants, to ensure the effectiveness of the agroinfiltration. For plant species with thick and robust leaves, this technique of lightly incising the abaxial leaf surface provides accuracy, and avoids excessive tissue damage. Since whole plant agroinfiltration requires a properly equipped greenhouse, or walk-in growth chamber, detached leaves placed in floral foam and plastic containers require less space, and enable agroinfiltration in a laboratory environment with an appropriate safety level.

Using genetic markers and effector gene agroinfiltration, we successfully obtained ten *R8* potato genotypes, selected from the progeny of five susceptible parents and the resistant Sárpo Mira.

### 3.2. Slovenian Isolate 02_07 Was the Least Aggressive Isolate despite Belonging to EU_13_A2 Genotype

After obtaining the final collection, the *R8* genotypes were inoculated with four *P. infestans* isolates, along with their parental cultivars, in order to evaluate their resistance level and compare them to their corresponding parental cultivar. The four isolates differed in their aggressiveness, geographical origin, and growth. The two foreign complex isolates, IPO-C and 90128, are known for their aggressiveness, and are frequently used in potato late blight resistance studies [24,31,46,51,54,56,59]. In this study, the aggressiveness of the two Slovenian *P. infestans* isolates, 02_07 and 09_07, was determined and reported for the first time. The isolate 02_07 was the least aggressive, while isolate 09_07 was intermediate, ranked by a Tukey’s range test as in between 02_07 and the foreign isolates IPO-C and 90128 (Figure 4). The isolates 02_07 and 09_07 were isolated from infected potato leaves in Slovenian fields in 2007 [60]. As part of this population study, both isolates were tested for mating type and metalaxyl resistance, and genotyped with microsatellite markers. The isolate 02_07 belongs to mating type A2, and is resistant to metalaxyl, while the isolate 09_07 belongs to mating type A1, and is susceptible to metalaxyl. Microsatellite analysis of twelve SSR markers reveals that the isolate 02_07 belongs to genotype EU_13_A2, which is the aggressive genotype that dominated western European potato fields from 2004 to 2017 [14]. Although several studies that use EU_13_A2 isolates for late blight infection confirm this aggressiveness [35,38,61], this was not the case in our study. Among all the progeny *R8* genotypes and parental cultivars tested in this study, only the cultivar Rioja shows some late blight symptoms after inoculation with the isolate 02_07 (Table 1, Appendix A), and in general, 02_07 rarely produced sporangia after mycelial emergence, which is an indication of a successful infection. In a 2015 study by Mariette et al. [62], several *P. infestans* EU_13_A2 isolates are tested for aggressiveness and invasiveness. The EU_13_A2 isolates produce significantly fewer spores, and have the lowest sporulation capacity among tested isolates; however EU_13_A2 isolates collected in 2004–2005 cause the largest lesions, while the EU_13_A2 isolates collected in 2006–2008 cause the smallest lesions. These data are in partial agreement with our results, since the isolate 02_07 used in our study produced few sporangia, even on susceptible cultivars. However, as we did not measure late blight lesion sizes, we cannot assess the ability of lesion formation. It was suggested that these discrepancies were due to phenotypic variability within the same clonal lineage, which is characteristic of genetically distinct isolates [62]. One of the suggested possibilities for this difference was the geographical origin of the collected isolates, as different locations and their climate conditions could influence the virulence phenotype. Further studies, such as determining the race of the isolate 02_07 by infecting Mastenbroek differential set [63], are needed to fully test and evaluate the virulence and aggressiveness of 02_07. The second Slovenian isolate, 09_07, was classified as genotype EU_34_A1, which was predominant in Poland in 2008 [64]. Although the AUDPC of the isolate 09_07 was intermediate, between that of 02_07 and the foreign isolates 90128 and IPO-C, it has a high sporulation ability, similar to IPO-C, and could be described as aggressive.

Sárpo Mira was used in late blight resistance studies, where it was inoculated with isolate EU_13_A3, and it shows very mild, or no, late blight symptoms [35,38], which is in correspondence to our results, since Sárpo Mira is also resistant to the EU_13_A2 genotype in our study. However, since the aggressiveness of the EU_13_A2 genotype is not consistent in other resistance studies, and all susceptible parental cultivars in our study show similar resistance to this isolate, we conclude that the isolate 02_07 is not aggressive, despite belonging to the genotype EU_13_A2. 

### 3.3. Rpi-Smira2/R8 Gene Is Capable of Conferring Resistance to Late Blight Comparable to Sárpo Mira, Although This Trait Depends on the Genetic Background

The progeny *R8* genotypes differed in their level of resistance depending on the inoculated *P. infestans* isolate. All cross groups are resistant to the isolate 02_07, while the disease severity for the other three isolates (09_07, 90128, and IPO-C) is higher in all four cross groups, showing an intermediate or susceptible phenotype.

The genotype L166, from the Lusa cross group, is the only progeny *R8* genotype with higher MD scores compared to its susceptible parental cultivar after inoculation with the isolates 02_07, 09_07, and 90128, although this difference is only statistically significant for the least aggressive isolate 02_07.

The progeny genotypes from the Rioja cross group (R7 and R15) do not statistically significantly differ from their parental cultivar Rioja after inoculation with the isolates 09_07, 90128, and IPO-C, showing similar intermediate or susceptible response to these *P. infestans* isolates. Analysis with genetic markers for the genes *R3a* and *R3b* reveals the presence of both minor genes in the cultivar Rioja (Figure 3), which could potentially affect the disease response of this cultivar to late blight in this study. Although the cultivar Rioja has lower MD scores, and higher resistance, compared to the cultivars Sylvana and Colomba after inoculation with the isolates 09_07, 90128, and IPO-C, Rioja still has higher MD scores, and shows more late blight symptoms compared to the cultivar Lusa (Table 1), which is shown to have none of the four tested *R* genes (Figure 3). This indicates that the *R3a* and *R3b* genes have minor, or no, effect on late blight resistance in the cultivar Rioja.

In some cases, the progeny *R8* genotypes from the Sylvana and Colomba cross groups show statistically significantly higher resistance levels compared to the parental cultivars Sylvana and Colomba, respectively. The genotypes S989 and S999 are intermediately resistant to the isolate 09_07, but are more susceptible to this isolate when compared to Sárpo Mira. They show no statistically significant difference to Sylvana after infection with the isolates 90128 and IPO-C (Table 1). Although the cultivar Sylvana tested positive for the minor *R3b* gene (Figure 3), it is still highly susceptible to the isolates 09_07, 90128, and IPO-C, along with the cultivar Colomba, which indicates little or no effect of the *R3b* gene in the cultivar Sylvana. The genotype C571 has the lowest MD scores among the progeny *R8* genotypes after inoculation with the isolates 09_07, 90128, and IPO-C, and is statistically significantly more resistant to late blight compared to the parental cultivar Colomba (Table 1). In the case of the isolate IPO-C, there is no statistically significant difference between the genotype C571 and Sárpo Mira, indicating a similarly intermediate late blight response between the progeny *R8* genotype and the resistant parental cultivar. However, this similarity is more likely due to Sárpo Mira having a relatively high MD score for this isolate, even though Sárpo Mira is shown to be resistant to the isolate IPO-C [39,51,65]; therefore; ANOVA and post-hoc analysis ranks the genotype C571 and resistant cultivar in the same group. Nevertheless, both the genotype C571 and Sárpo Mira have intermediate late blight response to the isolate IPO-C when compared to the genotypes C419, C557, and parental cultivar Colomba, which are all highly susceptible. In addition, as mentioned earlier, the genotype C571 has the lowest MD scores for the isolates 09_07, 90128, and IPO-C among the nine tested *R8* genotypes. 

These results suggest that the *R8* gene of Sárpo Mira contributes to the qualitative resistance of the progeny *R8* genotypes. This resistance, however, is not the same for all *R8* genotypes used in this study, even if they contain the same major *R* gene. Studies by Vossen et al. [50] and Kim et al. [51] suggest that the level of *R8* gene resistance is highly dependent on the genetic background, which may explain the differences in resistance levels among the tested *R8* genotypes. Both studies use *R8* plants obtained either by sexual crossing [51] or by transgenesis [50], which report similar levels of resistance to that of Sárpo Mira under field conditions. However, in both studies, this high level of resistance is lost either during detached leaf assay (DLA), or under climate chamber conditions. This is consistent with other studies in which resistance of Sárpo Mira to late blight varies depends on whether whole plants or detached leaves are infected [40,41,42]. Detached leaves are shown to be more susceptible to *P. infestans* compared to whole plants or field trials, due to different experimental conditions [31,66,67]. In addition, intermediately aggressive *P. infestans* isolates are more suitable for this type of late blight resistance study, due to their characteristic logarithmic growth on the host plant (Figure 5b and Appendix A). This type of disease progression reveals the minor response differences between the tested potato genotypes, and helps to better evaluate the contribution of individual *R* genes compared to control plants, such as susceptible parental cultivars, as was the case in our study. The use of the overly aggressive isolates 90128 and IPO-C in our study masks these differences between the progeny *R8* genotypes, since their disease progression resembles linear growth (Figure 5c,d and Appendix A), and makes it difficult to distinguish whether the tested progeny *R8* genotypes are more susceptible to these isolates due to the poor contribution of the *Rpi-Smira/R8* gene to combating late blight, or if the immune system is simply overwhelmed by the aggressiveness of the pathogen.

Despite previous reports of Sárpo Mira’s high resistance to the isolate IPO-C [39,51,65] this is not completely the case in our study, since the isolate IPO-C sporulates on some Sárpo Mira plantlets, which is an indication of a successful infection with *P. infestans*. This is probably due to the in vitro conditions of the experiment. Microbe-free plant material, combined with controlled conditions in the growth chamber (optimal temperature and high humidity), present favorable conditions for symptom development and the growth of *P. infestans*. Moreover, plant tissue cultures cannot be compared equally to greenhouse or field-grown whole plants, therefore, some differences in disease development are to be expected. In the study by Orłowska et al. [42], a signaling pathway is induced by late blight infection, where plant roots and meristems play an important role. Since root and meristem development might be altered in in vitro plantlets, this could influence the extent of resistance to late blight. 

Overall, experiments on late blight infection in vitro is a suitable method for molecular and disease development studies related to plant-pathogen interaction, as in vitro plantlets are uniform and provide optimal conditions for a successful inoculation, with no interference of other pathogens [53], which can occur in detached leaf assays. Additionally, plantlets in tissue culture are easily examined under stereomicroscope, without disrupting the disease progression, and no destruction of the plant material. This allows for accurate monitoring of late blight symptom development and detailed observation of even minor changes among tested potato genotypes, such as the emergence of mycelia and presence of sporangia.

## 4. Materials and Methods

### 4.1. Plant Material

A total of 1420 true seeds were acquired from the potato breeding program at the Agricultural Institute of Slovenia, where five susceptible cultivars (Rioja, Lusa, Bikini, Colomba, and Sylvana) were crossed with the resistant cultivar Sárpo Mira. In November 2017, true seeds were sown in soil-filled trays in the greenhouse (growing conditions: 16 h/8 h day/night regime, temperature 21 ± 3 °C). After four weeks, a total of 1213 plantlets reached full germination, and were planted in individual soil-filled pots (Figure 1).

During the selection process, plants were clonally propagated using cuttings with either apical (shoot), or intercalary meristem (nodes from second to fourth leaves, from six to eight week old plants). Each cutting, containing one upper node with leaf and one lower node with leaf removed, was dipped in paste containing the auxin hormone naphthalene acetic acid (NAA) and activated charcoal (0.1 g/mL). Cutting paste was prepared by dissolving 30 mg NAA in a few drops of 1 M NaOH, then adding distilled water up to 100 mL, and gradually adding 10 g activated charcoal to obtain a thick consistency. The dipped cuttings were placed into sowing trays in the greenhouse, and covered with clear plastic film to maintain high humidity until root formation. After three weeks, the cuttings were transferred to soil-filled pots. 

After obtaining the final plant collection (ten progeny genotypes and five parental cultivars), the single nodal stem explants (approximately 1 cm in size) of the greenhouse-grown potato plants were sterilized with 0.5% (*m*/*v*) IZOSAN G (PLIVA, Zagreb, Croatia) for 10 min, washed with sterile water three times, and transferred to Murashige and Skoog (MS, supplemented with 30 g/L sucrose and 8 g/L agar at pH 5.7 [68]) medium for in vitro micropropagation (Figure 1). In vitro plantlets were maintained and sub-cultured every four weeks in glass jars (10 cm diameter) with 25 mL of fresh MS medium. Cultures were grown at 22/20 °C under a photoperiod of 16 h light day, with a light intensity of 3200 lux. 

### 4.2. Marker-Assisted Selection (MAS) of Potato Genotypes

For selection with genetic markers, DNA was extracted from 4 week old greenhouse grown plants using the Biosprint15 DNA Plant Kit (QIAGEN, Hilden, Germany) and the MagMax Express Magnetic Particle Processor (Life Technologies, Carlsbrand, CA, USA), according to the manufacturer’s instructions. The quality of genomic DNA was determined with electrophoresis on a 1.4% agarose gel. The oligonucleotides and conditions used for PCR amplification of the qualitative resistance genes *R3a*, *R3b*, *Rpi-Smira1,* and the quantitative resistance gene *Rpi-Smira2/R8*, as well as the expected amplicon length, are listed in Table 2. The PCR protocol was as follows: initial denaturation 4 min at 95 °C, 35 cycles of 30 s denaturation at 95 °C, 30 s at annealing temperature (Ta in Table 2), 1 min extension at 72 °C, and 4 min final extension at 72 °C. PCR reactions were performed using the Kapa3G Plant PCR Kit (Sigma-Aldrich, St. Louis, MI, USA) and Veriti™ thermal cycler (Applied Biosystems, Waltham, MA, USA). Potato genotypes were selected in several steps (Figure 1). First, PCR amplicons of the *R3b* gene were verified in agarose electrophoresis, and only negative samples were used for subsequent selection of plants with markers for the *R3a* and *Rpi-Smira1* genes. Finally, the R8-184 marker was used to obtain a population of plants carrying the *R8* gene. The *R8* fragments were cleaved using the restriction enzyme RsaI (New England Biolabs, Ipswich, MA, USA), according to the manufacturer’s instructions. Positive plants were used for further screening and negative selection for the *R4* gene.

### 4.3. Selection of R4-Negative Plants Using Avr4 Effector Gene Agroinfiltration

The presence of the *R4* gene was determined using agroinfiltration of the *Avr4* effector gene. *Agrobacterium tumefaciens* strain Agl1 + VirG, containing vector pK7WG2 with the *Avr4* gene (PITG_07387), was kindly provided by Francine Govers (Laboratory of Phytopathology, Department of Plant Sciences, Wageningen University and Research). Agroinfiltration was performed according to Van Poppel et al. [24], with slight modifications. Briefly, *A. tumefaciens* was grown overnight in 50 mL YEB media (5 g beef extract, 5 g bacto-tryptone, 5 g sucrose, and 1 g yeast extract per liter [52]), supplemented with 200 µM acetosyringone, 10 µM MES (2-morpholinoethanesulfonic acid), and the antibiotics spectinomycin 100 µg/mL and rifampicin 25 µg/mL. When the optical density at 600 nm (OD_600_) reached 0.8, cells were centrifuged and resuspended in an infiltration buffer consisting of 10 mM MES, 10 mM MgCl2, and 200 µM acetosyringone. The resuspended bacteria were incubated in the dark at room temperature for one hour. Detached leaves from 3 week old greenhouse-grown plants were first nicked several times with a needle on the abaxial side under a stereomicroscope, and then syringe-infiltrated with the bacterial suspension. *A. tumefaciens* with empty pK7WG2 vectors and sterile water were used as a mock control and negative control, respectively. After three days, leaves were examined for the hypersensitive response indicating the presence of the *R4* gene (Figure 2).

### 4.4. Maintenance of Phytophthora infestans Isolates and Inoculum Preparation

Four different *P. infestans* isolates were used in this study (Table 3). The isolates IPO-C (race 1.2.3a.3b.4.5.6.7.10.11) and 90128 (race 1.3a.3b.4.6.7.8.10.11) were kindly provided by Trudy van den Bosch (Biointeractions and Plant Health, Wageningen Plant Research, Wageningen University, and Research). The isolates 02_07 and 09_07 are classified as genotype EU_13_A2 and EU_34_A1, respectively, although their virulence is unknown. *P. infestans* isolates were cultured on Rye-A medium, [69] supplemented with 20 g/L sucrose and incubated in the dark at 20 °C. Mycelial plugs were transferred to fresh Rye-A medium every two weeks. 

To ensure the virulence of the pathogen [70], all *P. infestans* isolates were passed through susceptible potato cultivar Dobrin [71] (Figure 6). Sporangia suspension was prepared by washing the mycelia-covered Rye-A plates with sterile water and scraping the sporangia with a metal spatula. Sporangia suspension was incubated in the dark at 6 °C for two hours, to induce the release of zoospores. After incubation, hyphal fragments were removed by filtering the suspension with 100 µm cell strainers (Corning^®^, Sigma-Aldrich, St. Louis, MI, USA), then transferred to 50 mL glass spray bottles (Figure 6a). Glass jars with four to five week old in vitro plantlets of Dobrin were evenly sprayed with zoospore suspension. After five to six days, mycelium-covered leaflets were transferred to Rye-A+ medium (Figure 6b) supplemented with 20 g/L sucrose, antibiotics ampicillin (250 mg/L), rifampicin (10 mg/L), and antimycotic pimaricin (10 mg/L). The plates were incubated in the dark at 20 °C for three weeks, until sporangia formation.

Mycelium-covered Rye-A+ plates from infected Dobrin plantlets were then used to prepare the inoculum for the late blight resistance assays. The preparation protocol was similar to that described for Dobrin. After the Rye-A+ plates were washed and scraped, an aliquot of the suspension was taken to count the sporangia with a hemocytometer (Neubauer Improved, Glaswarenfabrik Karl Hecht GmbH, Sondheim vor der Rhön, Germany). The concentration was adjusted to 2.5 × 10^4^ sporangia/mL, and 0.01% (*v*/*v*) Tween 20 was added. The sporangia suspension was incubated in the dark at 6 °C for two hours, filtered with 100 µm cell strainers (Corning^®^, Sigma-Aldrich, St. Louis, MI, USA), and transferred to 50 mL glass spray bottles (Figure 1).

### 4.5. In Vitro Inoculation of R8 Plants with Phytophthora infestans Zoospores

Four to five week old in vitro *R8* plantlets were used for the experiment. Glass jars containing three plantlets were evenly spray-inoculated with zoospore suspensions. Control plantlets were spray-inoculated with sterile water, supplemented with 0.01% Tween 20. The plastic lids of the jars were covered with parafilm to maintain high humidity. The glass jars were placed in a growth chamber (14 h/10 h day/night regime at 22 °C and 75% relative humidity). Middle, fully developed leaves of inoculated plantlets were visually inspected under a stereomicroscope daily for eight days for symptom development and pathogen growth. Each plantlet was given a disease score ranging from 0 (no symptoms) to 8 (leaves covered with mycelia and sporangia), depending on the severity of symptoms on the leaves (Figure 7). Representative leaflets with late blight symptoms were photographed (Figure 7) using a digital microscope (Olympus DSX1000, objective DSX10-SXLOB1X, working distance 51.7 mm). ImageJ software [72] was used to remove the background of the images and set to black.

Progeny *R8* genotypes and parental cultivars were inoculated separately with Slovenian *P. infestans* isolates (02_07, 09_07) and foreign isolates (90128, IPO-C). Each inoculation experiment with one of the two groups of *P. infestans* isolates was independently repeated twice. 

### 4.6. Statistical Analysis

In this study, six glass jars containing three plantlets (eighteen plantlets total) per genotype or cultivar, from two replicated inoculation experiments with either Slovenian or foreign *P. infestans* isolates, were examined daily for late blight symptoms. Contaminated glass jars were removed from the experiment, leaving fifteen inoculated plantlets for genotype R15 infected with isolates 02_07 and 09_07, genotype S999 infected with isolate 02_07, genotype L166 infected with isolate 90128, and cultivar Sylvana infected with isolate IPO-C. 

The mean disease (MD) score was calculated for each inoculated genotype or cultivar as the mean of the scores for eighteen (or fifteen) plantlets, and the resulting data were divided into four groups, according to progeny and corresponding susceptible parental cultivar. The genotype or cultivar was considered resistant if the MD score at eight days post inoculation (dpi) was below 3.9, intermediate if the MD score was between 4.0 and 5.9, and susceptible if the MD score was above 6.0. One-way analysis of variance (ANOVA) was applied only to the data sets at 8 dpi to compare each progeny *R8* genotype with the corresponding susceptible and resistant parental cultivar. The threshold for statistical significance was *p* < 0.05, and Tukey’s range test was used for post-hoc analysis.

The area under the disease progress curve (AUDPC) for each *P. infestans* isolate was calculated using trapezoidal rule [73], and one-way ANOVA was used to compare disease progression and aggressiveness in all potato plantlets tested. Tukey’s range test was also used for post-hoc analysis.

All statistical analyses were performed using R Studio software [74] (version 4.0.3), package “agricolae”.

## 5. Conclusions

In this study, we successfully obtained ten *R8* genotypes with only one resistance gene (*Rpi-smira/R8*) from the resistant parent Sárpo Mira by using genetic markers and effector agroinfiltration. Nine of these genotypes were inoculated with two Slovenian and two foreign *P. infestans* isolates, and we confirm the high aggressiveness of the isolates 90128 and IPO-C, while 09_07 is moderately aggressive. The isolate 02_07, although identified as genotype EU_13_A2, is the least virulent. 

The progeny *R8* genotypes of the Colomba and Sylvana cross groups show statistically significantly milder symptoms compared to the susceptible parental cultivar, with the genotype C571 being the most resistant among all nine progeny genotypes to the moderately and highly aggressive *P. infestans* isolates used in this study. These results suggest that the *Rpi-Smira2/R8* gene confers resistance to late blight, but is not the most contributive gene, as previously considered. In addition, the presence of this gene alone does not ensure high resistance, and is influenced by the genetic background, since the progeny *R8* genotypes differ in late blight response. 

Although in vitro experiments performed in this study provided a contamination-free environment and optimal infection conditions, field trials or whole plant experiments need to be conducted in parallel to confirm the level of resistance shown in the *Rpi-Smira2/R8* genotypes. To fully understand the influence of the genetic background of the *Rpi-Smira2/R8* gene, and its contribution to late blight resistance, further studies, such as gene expression studies, coupled with genome sequencing of the progeny *R8* genotypes and Sárpo Mira, are required. Additionally, effectoromics could be applied to the progeny *R8* genotypes to confirm proper functioning of the *Rpi-Smira2/R8* gene. Furthermore, progeny genotypes containing two or more *R* genes from Sárpo Mira could be selected and tested for late blight resistance, in order to identify potential *R* gene interactions. 

## Figures and Tables

**Figure 1 plants-11-01319-f001:**
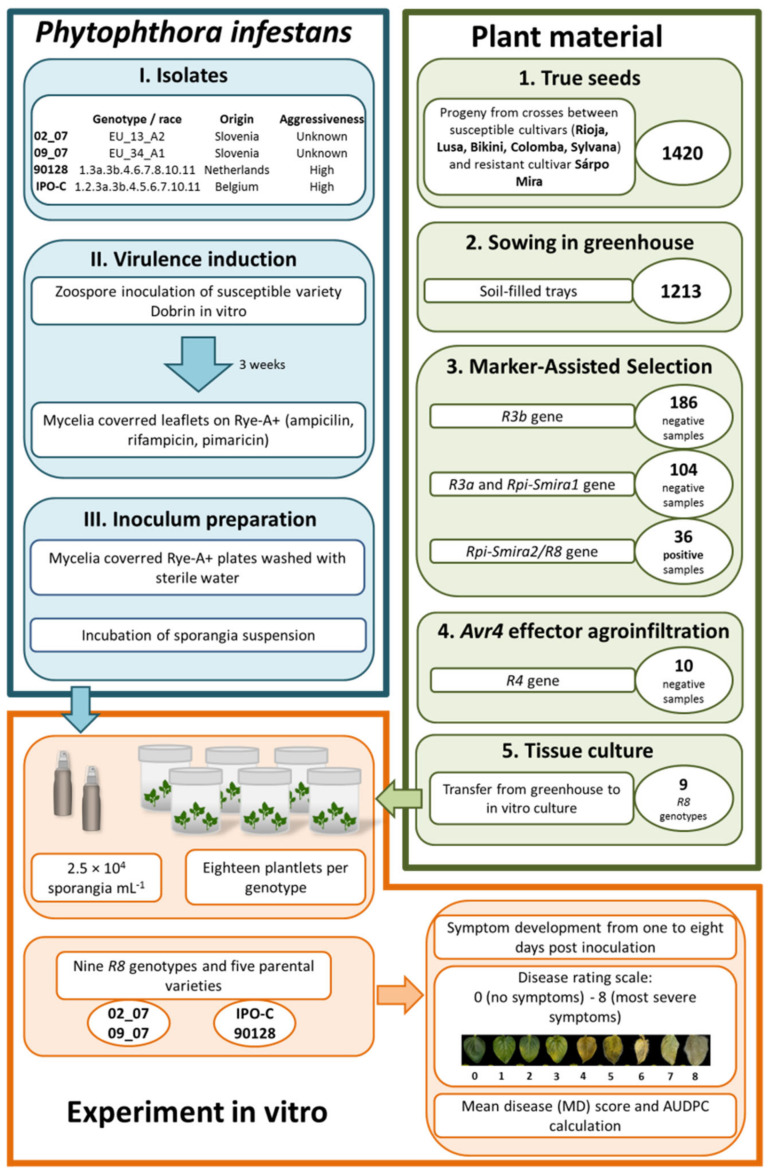
Diagram of *Phytophthora infestans* inoculum preparation, selection of *R8* potato genotypes, and experimental design of *P. infestans* inoculation of *R8* genotypes in vitro.

**Figure 2 plants-11-01319-f002:**
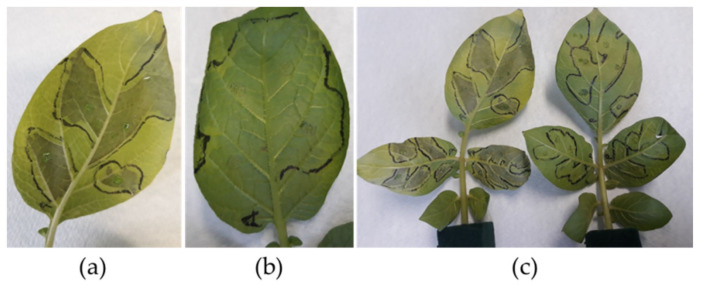
The *Avr4* effector agroinfiltration discriminates potato plants with (**a**) or without (**b**) *R4* gene. (**a**) Leaflets displaying hypersensitive response after agroinfiltration of *Avr4* effector gene. (**b**) Leaflets without *R4* gene showing no response after agroinfiltration of *Avr4* effector gene. (**c**) Leaves with hypersensitive response (**left**) compared to leaves agroinfiltrated with empty vector, mock control (**right**).

**Figure 3 plants-11-01319-f003:**
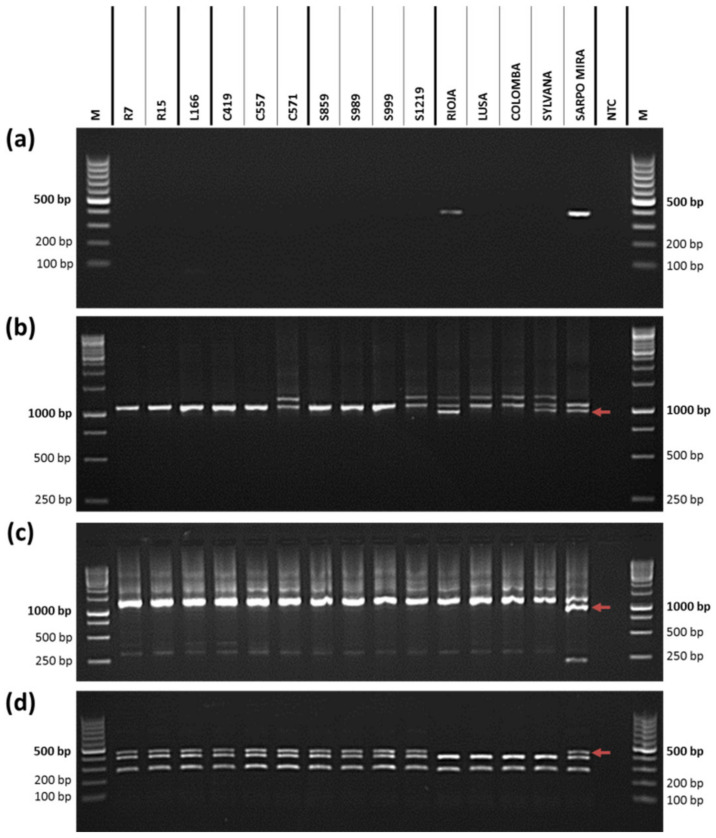
Absence of the (**a**) *R3b* gene, (**b**) *R3a* gene, (**c**) *Rpi-Smira1* gene, and presence of the (**d**) *Rpi-Smira2/R8* gene in the final collection of ten *R8* progeny genotypes and parental cultivars. Progeny *R8* genotypes are listed as follows: R7, R15 (Rioja cross); L166 (Lusa cross); C419, C557, and C571 (Colomba cross); S859, S989, S999, and S1219 (Sylvana cross), separated by bold lines; (**a**) only Rioja and Sárpo Mira are positive for marker R3b (378 bp), and contain the *R3b* gene; (**b**) Rioja, Sylvana, and Sárpo Mira are positive for marker Sha (the red arrow marks the amplicon length of 982 bp), and contain the *R3a* gene; (**c**) only Sárpo Mira is positive for marker 45/XI (the red arrow marks the amplicon length of 1000 bp), and contains the *Rpi-Smira1* gene; (**d**) ten *R8* progeny genotypes and Sárpo Mira are positive for marker 184-81 (the red arrow marks the amplicon length of 480 bp) after restriction with RsaI, and contain the *Rpi-Smira2/R8* gene. M represents 100 bp ladder in case of R3b (**a**) and 184-81 (**d**) marker, and 1000 bp ladder in case of Sha (**b**) and 45/XI (**c**) marker. NTC stands for non-template control.

**Figure 4 plants-11-01319-f004:**
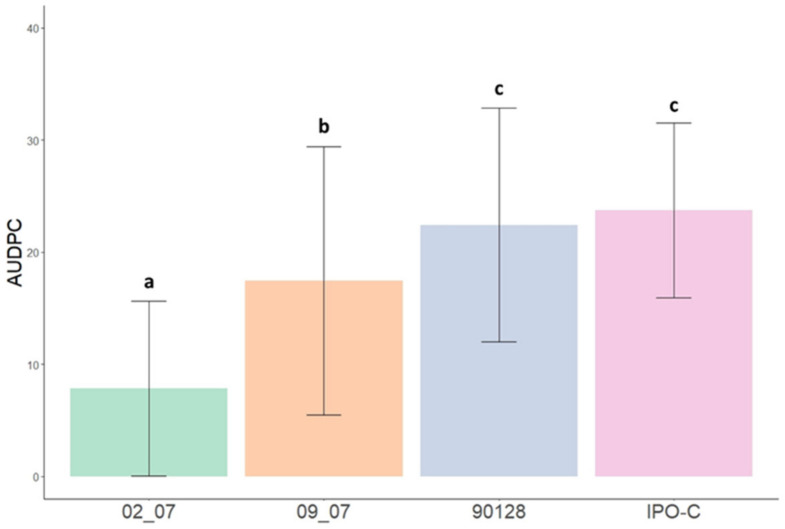
Area under the disease progression curve (AUDPC) values with standard deviation of each *Phytophthora infestans* isolates (02_07, 09_07, 90128, and IPO-C). ANOVA and Tukey’s analysis (*p* < 0.05) were used to determine the level of aggressiveness of each isolate across all tested potato genotypes and parental cultivars (*n* = 14). Same letters for individual *P. infestans* isolate indicate no statistical significance.

**Figure 5 plants-11-01319-f005:**
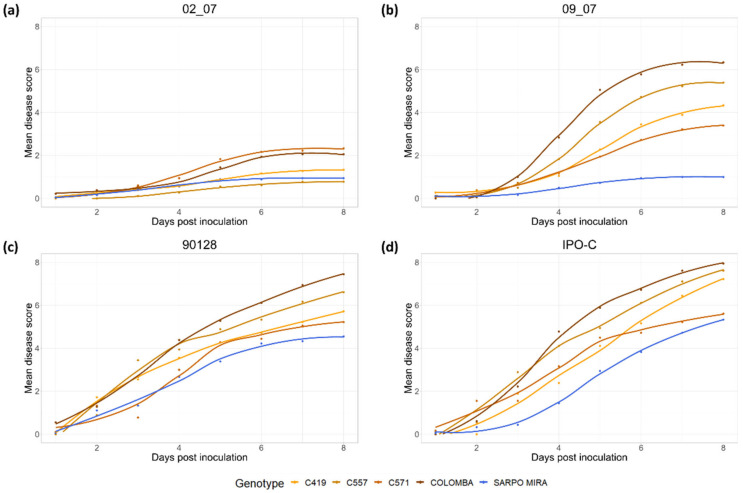
Late blight disease progression curves of the Colomba cross group show intermediate resistance of the progeny *R8* genotypes compared to parental potato cultivars, after inoculation with *Phytophthora infestans* isolate 02_07 (**a**), isolate 09_07 (**b**), isolate 90128 (**c**), and isolate IPO-C (**d**). Genotype C571 is the only progeny that shows comparable disease resistance to Sárpo Mira, both after inoculation with isolate IPO-C and 90128. Genotypes C419 and C557 show milder symptoms compared to the susceptible parental cultivar Colomba, after inoculation with isolates 09_07, 90128, and IPO-C. All three genotypes and parental cultivars are resistant to isolate 02_07.

**Figure 6 plants-11-01319-f006:**
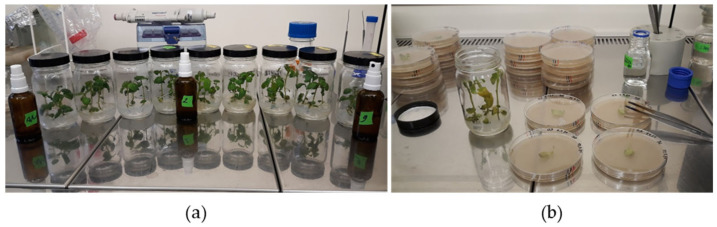
Inoculation of *Phytophthora infestans* on susceptible potato cultivar Dobrin in vitro. (**a**) Four to five week old in vitro plantlets of Dobrin were used to maintain virulence of *P. infestans* isolates before experiments. Zoospore suspension was inoculated using 50 mL glass spray bottles. (**b**) Five to six days post inoculation mycelium-covered leaflets were transferred to Rye-A+ medium.

**Figure 7 plants-11-01319-f007:**
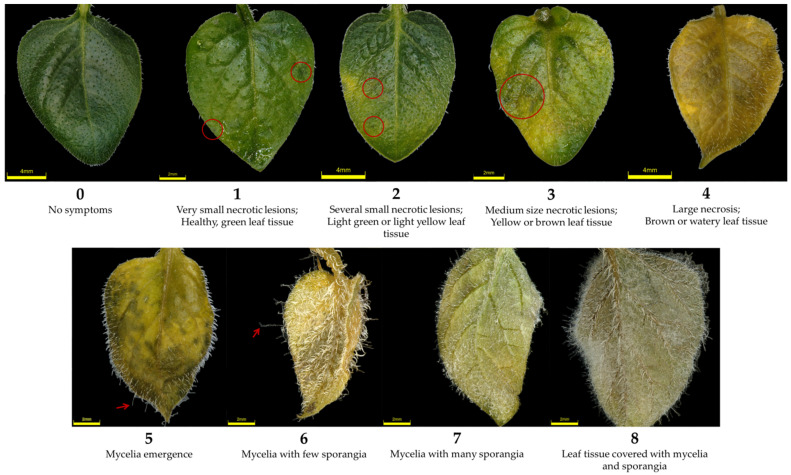
Disease rating scale used for determining the severity of late blight after inoculation of *R8* genotypes with *Phytophthora infestans* with representative leaflets for each disease score. Necrotic lesions are indicated with red circles, while mycelia and sporangia are marked with red arrows. All images were taken with darkfield illumination and 20× magnification.

**Table 1 plants-11-01319-t001:** Mean disease (MD) scores for progeny *R8* genotypes divided into four cross groups, after infection with each *Phytophthora infestans* isolate (02_07, 09_07, 90128, and IPO-C), at 8 dpi. Each cross group consists of MD scores for the progeny *R8* genotypes, the corresponding susceptible parental cultivar, and the resistant cultivar Sárpo Mira. ANOVA was performed within each cross group, to compare the progeny *R8* genotypes with the parental cultivars for each *P. infestans* isolate, and Tukey’s range test was used for post-hoc analysis. Same letters within the cross group for individual *P. infestans* isolate indicate no statistical significance.

	*Phytophthora infestans* Isolates			
	02_07	09_07	90128	IPO-C
Rioja	3.28 ^a^	5.83 ^a^	6.00 ^ab^	5.94 ^ab^
R7	2.56 ^ab^	6.00 ^a^	5.67 ^ab^	7.39 ^a^
R15	2.13 ^ab^	5.75 ^a^	6.50 ^a^	6.28 ^ab^
Sárpo Mira	0.94^b^	1.00 ^b^	4.56 ^b^	5.33 ^b^
Lusa	1.72 ^b^	4.89 ^a^	5.89 ^a^	6.82 ^a^
L166	3.33 ^a^	5.22 ^a^	6.44 ^a^	6.13 ^ab^
Sárpo Mira	0.94 ^b^	1.00 ^b^	4.56 ^b^	5.33 ^b^
Colomba	2.06 ^a^	6.33 ^a^	7.44 ^a^	7.94 ^a^
C419	1.33 ^a^	4.33 ^ab^	5.72 ^bc^	7.22 ^a^
C557	0.78 ^a^	5.39 ^ab^	6.61 ^ab^	7.61 ^a^
C571	2.33 ^a^	3.39 ^bc^	5.22 ^bc^	5.61 ^b^
Sárpo Mira	0.94 ^a^	1.00 ^c^	4.56 ^c^	5.33 ^b^
Sylvana	2.78 ^a^	7.17 ^a^	7.00 ^a^	7.40 ^a^
S859	1.56 ^a^	5.67 ^ab^	5.00 ^ab^	6.72 ^ab^
S989	1.83 ^a^	4.56 ^b^	5.89 ^ab^	7.06 ^ab^
S999	2.27 ^a^	4.11 ^b^	6.28 ^ab^	7.06 ^ab^
Sárpo Mira	0.94 ^a^	1.00 ^c^	4.56 ^b^	5.33 ^b^

**Table 2 plants-11-01319-t002:** Primers used for PCR amplification of genetic markers for plant selection of *R8* genotypes, annealing temperatures (Ta), and amplicon length.

Gene	Marker Name		Primer Sequence (5′-3′)	T_a_ (°C)	Amplicon Length (bp)	Reference
*R3a*	Sha	F	ATCGTTGTCATGCTATGAGATTGTT	60	982	[54]
		R	CTTCAAGGTAGTGGGCAGTATGCTT			
*R3b*	R3b	F	GTCGATGAATGCTATGTTTCTCGAGA	55	378	[51]
		R	ACCAGTTTCTTGCAATTCCAGATTG			
*Rpi-Smira1*	45/XI	F	AGAGAGGTTGTTTCCGATAGACC	58	1000	[36]
		R	TCGTTGTAGTTGTCATTCCACAC			
*Rpi-Smira2/R8*	184-81	F	CCACCGTATGCTCCGCCGTC	55	480 (RsaI)	[46]
		R	GTTCCACTTAGCCTTGTCTTGCTCA			

**Table 3 plants-11-01319-t003:** Characteristics of *Phytophthora infestans* isolates used in this study.

Isolate	Mating Type	Year	Genotype/Race	Origin	Reference
02_07	A2	2007	EU_13_A2	Slovenia	[60]
09_07	A1	2007	EU_34_A1	Slovenia	[60]
90128	A2	1990	1.3a.3b.4.6.7.8.10.11	Netherlands	[56]
IPO-C	A2	1982	1.2.3a.3b.4.5.6.7.10.11	Belgium	[34]

## Data Availability

Dataset supporting reported results are available at ECOBREED Zenodo community: https://doi.org/10.5281/zenodo.5521310.

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
