# Peer review of "Late Blight Resistance Conferred by *Rpi-Smira2/R8* in Potato Genotypes In Vitro Depends on the Genetic Background"

_plants, 2022, doi:10.3390/plants11101319_

Round 1

Reviewer 1 Report

When reviewing scientific papers for publication, I usually begin with a general overview in terms of structure, abstract, literature review, methodology, research findings, discussion, conclusions, and limitations of the paper.
The proposed paper is well written, and is devoted to a relevant scientific topic. The idea of the paper is well defined and its methodology is well proposed. The individual results are well discussed and justified. The final conclusion is adequate. However, there are also several problems that reduce the quality of the proposed work. The most important ones are as follows:

1.What knowledge gaps would you like to fill as an objective of this work? The presentation of technical research findings may not be adequate to fill the knowledge gaps in the field you are addressing. Please clearly address the main challenges, considering the research topic and your new contribution(s) to the field under discussion, as well as the context of your research.
2) I suggest expanding the list of literature studies to include 2020-2021. Enrich the literature review section with details of previous work in the relevant field and rely on those that clearly highlight the identified research gap and contributions. I recommend adding some references to the recent literature on the topic (including Web of Science and Scopus publications).
3. Suggest future scope of the study. How can the study be further expanded? Clearly address the main challenges, considering the research topic and your new contribution(s) to the field under discussion, as well as the context of your research.

Reviewer 2 Report

Dear Authors,

I have reviewed your manuscript "Late blight resistance conferred by Rpi-Smira2/R8 in potato genotypes in vitro depends on the genetic background" submitted for publication in Plants. It is a very interesting manuscript which is very well written and presented.

The manuscript provides new information on potato late blight resistance. The authors determined the contribution of the Rpi-Smira2/R8 gene to late blight resistance using in vitro potato progenies. Looking forward to seeing future research In field trials or in whole plant experiments to confirm the level of resistance of the Rpi-Smira2/R8 genotypes.

I suggest some edits that would improve even more the quality of it before publication. There are a number of points which the authors could consider (in the attached file).

I really recommend including pictures showing the in vitro inoculations (4.5) and the degree of  P. infestans infection in potato progenies. I suggest adding figure S1 as a figure in the manuscript rather than supplementary material. This can significantly improve the quality of the manuscript.

It is necessary to add references in some sentences in the introduction. I highlighted this in the attached document.

Describe the first time you mention an abbreviation; for example, line 207 dpi, line 488 (NAA), etc.

Add information about the size and age of cuttings used for vegetative propagation. Add the amount of activated charcoal used during the rooting procedure.

Carefully revised the format of the references. I highlighted this in the attached document.

For more specific comments, please see the reviewed manuscript attached.

Kind regards,
Reviewer
